# Simulation of Energy and Media Demand of Batch-Oriented Production Systems in the Beverage Industry

**Raik Martin Bär [1,\*], Sebastian Schmid [2], Michael Zeilmann [1], Joachim Kleinert [3], Karsten Beyer [3], Karl Glas [2] and Tobias Voigt [1]**

1 Brewing and Beverage Technology, Technical University of Munich, Weihenstephan, 85354 Freising, Germany; michael.zeilmann@tum.de (M.Z.); tobias.voigt@tum.de (T.V.)
2 Food Chemistry and Molecular Sensory Science, Technical University of Munich, Weihenstephan, 85354 Freising, Germany; sbstn.schmid@tum.de (S.S.); karl.glas@tum.de (K.G.)
3 SimPlan AG, 01217 Dresden, Germany; Joachim.Kleinert@simplan.de (J.K.); karsten.beyer@simplan.de (K.B.)
\* Correspondence: raik.baer@tum.de

**Abstract:** The global brewing industry is facing enormous environmental challenges and is urgently required to produce sustainably and efficiently. The rising costs of energy and electricity are forcing small and medium-sized breweries in particular, which are confronted with barriers such as lack of capital and know-how, to make substantial changes. This article presents an extended approach to prognose the energy and media demand for batch-oriented production of a brewery. Therefore, based on a modeling editor as well as a standardized data structure and an approach to determine the simulation-relevant parameters, a solution for fast and easy model generation was developed. Extensive measurement recordings within a brewhouse were performed to create a comprehensive model with recipe-specific parameters and detailed production plans. A simulation model can be generated automatically from a configuration file in a simulation environment that has been extended to include the mapping of batch-oriented operation. A validation is presented and a maximum deviation of the electrical and thermal energy demand of 1–2% is achieved. In combination with a preliminary work, the holistic simulation of the complex combined production of batch-oriented and discrete operation within the brewery is presented and allows comprehensive analysis as well as optimization towards sustainable production.

**Keywords:** modeling; simulation; energy and media demand; validation; brewing industry

## 1. Introduction

Not least due to the publication of the UN Sustainability Goals, companies are particularly required to produce sustainably [1]. The rising prices for energy and electricity are also forcing companies to act to continue to succeed in the market and to meet the demands of their customers. German companies are particularly affected by high energy and electricity prices [2]. The cost-driven food and beverage industry ranks third among the industrial sectors with the highest energy expenditures [3]. Beer production in particular is one of the most energy-intensive production processes in the food industry due to numerous heating and cooling processes in the brewhouse, numerous pumps and motors for material movement, and packaging and filling processes [4,5]. Large brewery groups are pursuing ambitious goals with regard to their sustainability, proclaiming targets such as $CO_2$ neutrality and halving water consumption by 2040 [6–8]. Thus, these efforts naturally also affect the supplying plant and mechanical engineering industry [9,10]. In particular, small and medium-sized companies (SMEs) in the beverage industry are urgently required to produce sustainably and energy-efficiently in order to withstand the enormous competitive pressure [11]. However, especially for SMEs, there are barriers

such as lack of capital, know-how, and data and information from production to achieve efficient and sustainable production [12]. IT-supported measures such as simulation analysis represent an opportunity to uncover optimization potential and can thus contribute to energy efficiency and sustainability. The advantages of simulation lie precisely in the representation of complex processes, which can also influence each other [13] and in the testing of possible measures by adapting the models without influencing or interrupting real production [14]. However, it is precisely the creation of the simulation model and the determination of the parameters required for the simulation which are the main challenges in simulations. In addition, simulation studies are usually associated with high costs and require a high level of expertise [15]. There are some approaches described in the scientific literature for using simulation to improve energy and media efficiency in breweries. However, these are limited in terms of some factors and barriers and allow only limited holistic analysis. In a previous paper, the authors describe a solution approach of modeling and automatic simulation model generation for the packaging or bottling area of the brewery and elaborate on the relevant literature [16].

Mignon and Hermia [17] describe for the first time the simulation of several brewhouses using the software BATCHES, a program for the simulation of batch and semi-continuous processes. The aim of the work is, on the one hand, to investigate the brewhouses with regard to their design and, on the other hand, to reduce the energy demand by adapting the process. Furthermore, using the same tool, it was determined how measures can contribute to reduce the peak loads of the steam consumption in the brewhouse [18]. Adjusting the production schedule can also contribute to reducing peak loads [19]. Muster-Slawitsch et al. [20] explain the importance and methods of correctly modeling the brewing process. Areas considered include the heat-intensive thermal processes of the brewhouse and the cooling load within the brewery. In both areas, real data obtained are compared to data modeled using mathematics. The simulation of a brewhouse using neural networks to predict energy consumption is described by Bai and Pu [21]. However, only a very small database is used here and no concrete application of the method is described. Other work deals with brewery waste and wastewater recovery using modeling and simulation [22–24]. Special attention must be paid to modeling using petri nets or so-called reference nets and simulation based on them. Hubert et al. [25] model and simulate the cooling demand within small and medium-sized breweries. Based on an extensive database with different recipes, a forecast can be made over a period of almost one year and a detailed validation be presented. Another approach regarding the simulative investigation of the refrigeration demand during fermentation using multi-agent systems is described by Howard et al. [26]. A holistic hybrid approach, which covers several areas of the brewery, is described by Hubert et al. [27]. They present a simulation, also based on reference nets and Java programming, in the form of balance flows including factors such as the chemical oxygen demand of wastewater and energy management. The application of the method is discussed in more detail by Hubert [28]. Here, simulations take place within a brewhouse, the cellar area, for CIP plants as well as in beverage bottling. However, the use cases mostly refer to the equipment module level and include numerous factors, e.g., with regard to the raw material or process quality or the design of the equipment. Nevertheless, the electrical power requirements of pumps and the water requirements for CIP cleaning are also optimized. Furthermore, simulations using recipe specificity as well as stored historical production plans show where the largest consumption of electrical and thermal energy and water as well as the production of wastewater occurs. The validation of the method is limited due to the lack of a comprehensive database to map the consumption behavior in detail. In addition, due to its complexity and the required expert knowledge, the method offers only a limited possibility for breweries, especially for SMEs.

The approaches mentioned have in common that no simultaneous consideration of several subareas of the brewery takes place and thus no reference is made to the different production methods. In addition, they are predominantly limited to one type of energy

and media. In particular, a recipe-specific consideration including a production plan takes place only in the works by Hubert et al. However, here, as in the other approaches, the database is described only to a limited extent or is not available and validation takes place only in part. Furthermore, a simple application of the presented methods is not given, since expert know-how or special software is required. These barriers pose particular challenges for SMEs in the brewing industry to increase their energy and media efficiency and to achieve sustainable production. A solution approach to overcome these challenges and barriers is presented by Bär et al. [16], but for the discrete packaging and filling area of the brewery. The present work takes up this approach and extends it to model and simulate the area of batch-oriented operation within the brewery with respect to its energy and media consumption.

This work presents an approach to the automatic generation of simulation models on the basis of an already presented modeling approach [29] and for the determination of simulation relevant parameters. Therefore, an extensive database was obtained by data measurement campaigns in the brewhouse of a medium-sized brewery. An overall model of the brewhouse with several recipes with specific parameter sets as well as with production schedules was created and, based on the configuration file of the modeling editor, a simulation model was automatically generated in a simulation environment, which was extended by the simulation of the batch-oriented operation. The simulation of the batch-oriented mode of operation is described in detail using the example of the brewhouse, and a detailed validation is presented. Furthermore, the holistic simulation of a complex production system consisting of the batch-oriented and discrete mode of operation is shown in the form of a use case by mapping the brewhouse and a beverage bottling plant in the simulation.

## 2. Brewhouse Investigated

Beverage production is divided into two fundamentally different production modes. First, the batch-oriented production of the beverage by means of process engineering measures must be mentioned [30,31]. In this processes, the raw materials required for the product are processed into the final product in several individual steps, with the steps influencing each other consecutively [31]. The ISA-88 standard regulates the subdivision of a batch process into its subprocesses and also describes an outline for the physical plants and recipes [32] and is described in detail for the application of a brewery by Bär and Voigt [29]. Figure 1 shows the brewing process with the respective units divided into the hot and cold blocks. The brewing process starts with the milling of the malt. The malt grains are mechanically broken up and mashed with water in the mash tun. The mash is then heated in individual temperature steps with specific durations. In some cases, a mash pan is also used for further thermal decomposition by boiling part of the mash. In the lauter tun, solid-liquid separation of spent grains and wort takes place. The obtained wort is boiled in the wort kettle with the addition of hops. After a whirlpool is used to separate the hot trub, the wort is cooled down to pitching temperature in a wort cooler [33]. This represents the interface to the cold block. The cold wort is transferred to a fermentation tank with the addition of yeast. The yeast starts alcoholic fermentation and breaks down the fermentable sugars into ethanol and $CO_2$. After fermentation, the young beer matures by storing it at low temperatures. Most beer types are filtered or centrifuged before the finished beer is put into a bottling tank prior to bottling [34]. Beverage filling represents the second part of the production modes due to the discrete mode of operation.

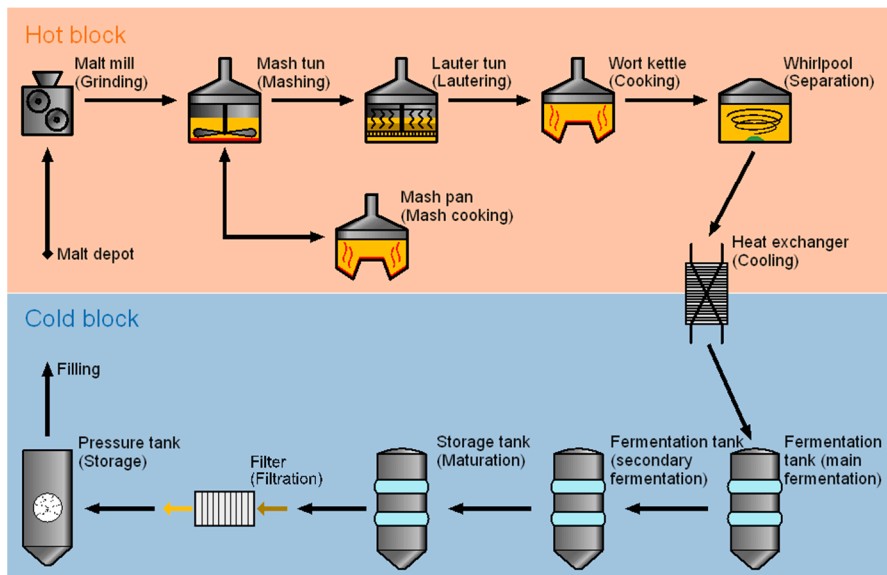

**Figure 1.** Brewing process divided into hot and cold blocks according to [34].

The brewery under investigation is a medium-sized German enterprise producing roughly 300,000 hectoliters per year of bottom- and top-fermented beers operating around the clock. The brewhouse presented in this paper has a cast-out wort volume of 100 hl per brew and, except for one special feature, corresponds to the structure presented. Regarding wort boiling, a gentle boiling system is used. In this process, the wort is kept hot only in the wort kettle and, after separation of the hot trub in the whirlpool, the wort is re-evaporated by means of a flash evaporator. The advantage is a high degree of flexibility in the boiling process and an increase in the quality of the wort [35]. Hot water (approx. 160 °C) is used as heat transfer medium throughout the brewhouse.

An overview of the consumption structures as well as the main consumers in the batch area of the brewery is given by Bär and Voigt [14]. It is important to note that the data described in the literature are in part subject to large fluctuations and are highly dependent on the size the brewery, its modernity, and the product portfolio [14,36,37]. The most intensive processes in terms of thermal energy are boiling and mashing in the brewhouse. Electrical energy is required in the brewhouse mainly for the operation of the grist mill, mash and wort pumps, and motors such as for the agitator of the mash tun and mash kettle and for the lauter tun raking machine [14]. The cooling of the tank and the pumping over of the beer require enormous amounts of electrical energy. Recuperative thermal energy is obtained and used in numerous processes. For example, the water heated in the countercurrent heat exchanger can be used for mashing the subsequent brew [34]. A uniform procedure for recording energy and media consumption in the brewery has not yet been described. Only this would enable reliable comparability. A standardized approach for mapping and modeling production systems, such as breweries, with regard to energy and media consumption is presented by Bär and Voigt [29].

### 3. Data Acquisition

To meet the data needs of the underlying modeling approach, we propose a technique [38] to assess the energy and media consumption of brewing equipment at the unit level [39] and related to the brewing process. The brewery investigated, like most small and medium-sized breweries [40], lacks the prerequisite necessary to quantify and qualify energy and media consumption at such a low level of aggregation. Thus, the method framework is designed to be set up for a limited period, providing the entire equipment required to obtain the default modeling parameters. Measurement points are defined at the equipment module level and the control module level for each unit of the process cell brewhouse. In this context, electric motors to drive pumps for instance,

constitute the equipment module level; the control module level is embodied by sensors (e.g., temperature sensors) and actuators (e.g., control valves). Where applicable additional data (e.g., recipe step sequences or motor switch signals) available from the brewhouse process control system (PCS) is used to derive the process model for each beer style and to qualify measurements, and are partly used to verify the measurements. To assess the energy consumption, suitable portable meters (PM) are selected for electrical energy (Janitza UMG 20CM channel branch circuit monitoring device; Janitza 20CM-CT6 operating and residual current monitoring module) and thermal energy (Flexim Fluxus F601 clamp-on energy flowmeter). Data from the PCS and PMs are recorded using a Windows-based computer running a Microsoft SQL Server database, with a resolution of up to 1 data point × s$^{-1}$ × parameter$^{-1}$. The PCS is connected via a local area network (LAN), whereas the PMs are connected via a programmable logic controller interface (PLC), as shown in Figure 2. The data format for a series of data points of a measurement point exported from database is as follows: measurement point ID—measured value—time stamp. The unit of each data point is retrieved from the PLC or the PCS. Correlation of the individual data points is facilitated employing a single system clock, even in the event of a non-equidistant recording.

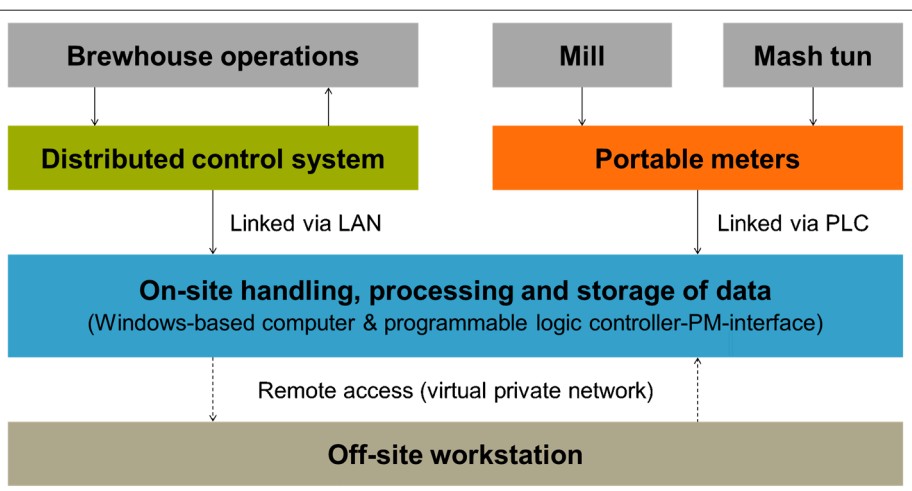

**Figure 2.** Quantification and qualification of energy and utility consumption at the unit level (according to [41]).

From milling the malt through to cooling the wort each unit of brewhouse was assessed, resulting in a total of 26 measurement points for electrical energy and a total of 6 measurement points for thermal energy. In addition, more than 100 parameters from sensors and actuators are incorporated. The measurements were divided into three consecutive series, each series covering a selected section of the brewhouse as well as energy types, and each covering a period of approximately 14–21 days).

MATLAB 2020b [42] software was used to prepare the raw data. During data import, the measured values were linearly interpolated over the entire measurement period of each data trace and normalized to equidistant time intervals of one second. The data were stored in an SQL database, which follows a generic and standardized structure as described by Bär et al. [16]. The data are divided into interval data and timestamp data. Interval data comprise the duration of a Process Operation of a unit, whereas timestamp data describe the consumption values at an exact point in time. The nomenclature and semantics of the data points correspond to the Weihenstephan Standards [43]. In addition, the database comprises tables containing units, data points and processes, subdivided up to the Process Operation. In addition, the production plan is imported. With the help of keys, it is possible to reference between the individual tables, which also enables easy

expansion of the database. This data structure is used for the raw data as well as for the simulated data.

## 4. Modeling Approach

To map the energy consumption behavior of a plant in the beverage industry completely and in detail, a context-free approach in the form of a metamodel was developed [29] and applied in this work. A metamodel, which describes models as a superordinate instance, is presented based on four modeling pillars. In addition to the physical model, the process model, a recipe or article model and a production plan model are defined. The modeling approach is implemented in a graphical modeling editor, which ensures fast, simple and structured model creation. The modeling approach has already been applied in detail for the packaging and bottling area within the beverage industry and described [29].

The physical model of the batch area (e.g., brewhouse) includes all real existing units, such as a mash tun, which are placed and linked as building blocks on a surface, described by a coordinate system. In the process model, the running processes and their sublevels (according to ANSI/ISA 88 [39]) are defined for the batch area (see Figure 3). Starting at the Process level, individual Process Operations are modeled via the Process Stage level. The Process Operation level represents the consumption-relevant level, in which they can be freely defined. The modeling of the process model in the modeling editor is based on the modeling language Business Process Model and Notation Language (BPMN), and four essential basic building blocks, which describe "Filling", "Emptying," "Processing" and "Setup", can be used. To define the sequence of the individual Process Operations, they are linked with arrows. The notation of BPMN makes it possible to depict the exact sequence of process steps across all three levels. The duration as well as the corresponding amount of energy and media required for the corresponding Process Operation are parameterized in the modeling editor. To simplify and speed up the modeling, required energy quantities can also be calculated. Factors can be used to describe the different energy and media efficiency of a unit as well as the share of recuperative energy.

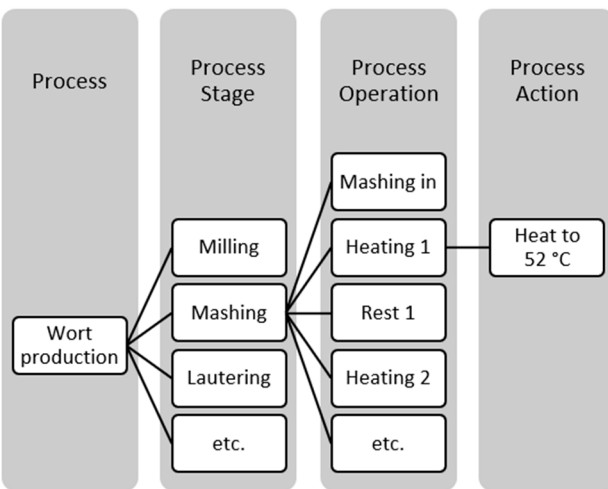

**Figure 3.** Division of the process model according to ANSI/ISA 88 [39].

Recipes are modeled by linking the Process Stages of a process model to units of the physical model, which increases the variability in the modeling and allows specific attention to be paid to the different product consumptions, but also to the differences in consumption of products in different Process Cells (e.g., brewhouse 1 and brewhouse 2). A process is modeled specifically for each recipe in the batch area. The production plan model covers all modeling columns with the definition of a flow chart, which contains the sequence of brews to be produced, as well as a matrix model for mapping the times

between the individual brews of different recipes (brew cycle). In addition, a shift plan is stored to limit the total available time. This sets the framework conditions for the later simulation of the model. The modeling editor generates an XML configuration file which contains all simulation-relevant parameters and structures.

## 5. Energy and Media Consumption Behavior and Determination of Simulation Parameters

The energy and media consumption behavior of batch-oriented processes is described in detail by Mignon and Hermia [17] and Muster-Slawitsch et al. [20]. The energy and media consumption depend on the currently applied process. According to the subdivision of ANSI-ISA 88 [39], the modeling approach uses the level of Process Operations in the process model as well as the unit level in the physical model. For example, considerably more thermal energy is required in a "Heating" Process Operation than in a "Resting" operation. With regard to electrical energy, clearly different consumption levels can be identified, e.g., pump or agitator operation. The following method is used to determine the individual Process Operations, their durations, and their energy and media requirements. The subdivision of the processes into individual Process Stages is determined on the basis of the units used. The division of a Process Stage (e.g., "Mashing") into Process Operations is based on the change in the consumption levels of the individual energy and media types. For this purpose, the data traces of the respective energy and media types of the individual consumers (equipment modules) of a unit or within a Process Stage are summarized and displayed graphically. The challenge in the definition of the Process Operations lies in the simultaneous consideration of the different energy and media types. If different consumption levels of different energy/media types overlap, they must be defined as further individual Process Operations. The start and end times of the Process Operations are derived manually and, in this case, stored in a SQL database following the presented data structure.

Figure 4 shows an excerpt of the course of the thermal and electrical energy consumption during the "Mashing" Process Stage. The Process Operations to be derived on the basis of the change in the respective power requirement can be seen in grayscale. The manual delimitation allows maximum and flexible granularity.

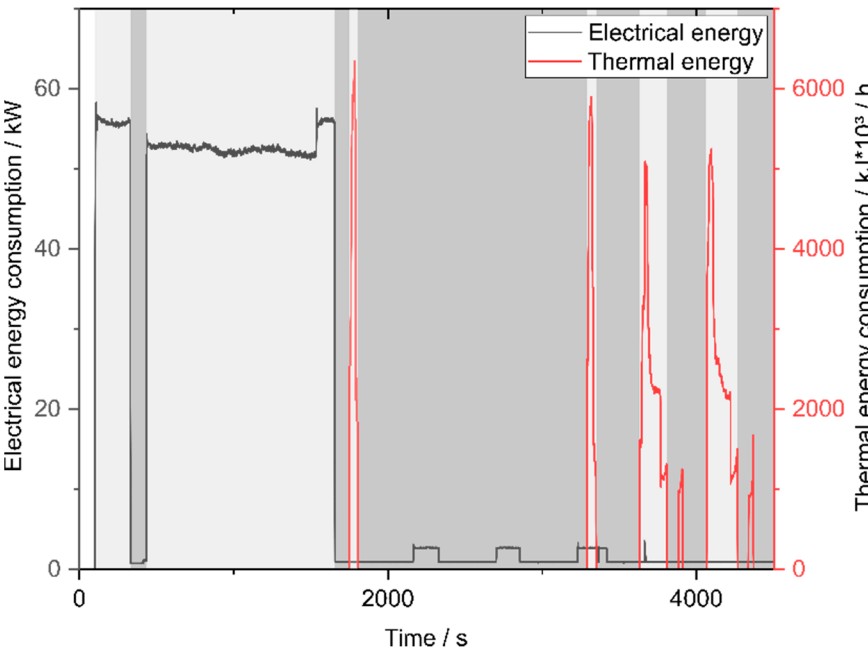

**Figure 4.** Example of the derivation of the individual Process Operations (highlighted in grayscale) based on the curves of the electrical and thermal energy demand for the Process Stages "Mashing" (section).

Based on the graphical plots of the electrical and thermal energy demand as well as the start and end times of the respective Process Operations, the required energy and media quantities can be determined by integration. Applications for this are common practice [44] and are used in various works [45,46].

The area under the data curve is determined using the trapezoidal rule (1). The numerical approximation of the integral of the function of the data curve is done by using several trapezoids of equal width. The width corresponds to one second in the present application.

$$\int_a^b y(x)dx = \frac{b-a}{2N}\sum_{n=1}^{N}(y(x_n) + y(x_{n+1}))\tag{1}$$

Since a unique consumption value $y$ is assigned to each second value $x$ in the data stored in the SQL database, the result is shown in (2).

$$\int_a^b y(x)dx = \frac{b-a}{2N}\sum_{n=1}^{N}(y(x_n) + y(x_{n+1})) = \frac{b-a}{2N}[y(x_1) + 2y(x_2) + \ldots + 2y(x_N) + y(x_{N+1})]\tag{2}$$

By using equidistant time intervals, energy can be determined by integrating power in the time intervals. An equivalent procedure is followed with respect to thermal energy. With a corresponding scope of data, the consumption values can be statistically examined and evaluated for each Process Operation on a recipe-specific basis. For this purpose, in particular, the automated determination of the consumption data based on the manually recorded times by means of software is recommended. The average values of the individual durations and the required amount of energy/media can then be used to parameterize a simulation model.

## 6. Simulation Environment

All simulation studies were carried out using the time-discrete "PacSi" simulation environment developed by SimPlan AG, Dresden, Germany. The simulation software masters the discrete simulation of packaging processes [47] and was further developed in the context of this work for the simulation of batch and thus continuous processes in the beverage industry with regard to their energy and media consumption. In combination with discrete manufacturing in the packaging sector, which has already been described comprehensively with regard to the mode of operation, validation and with simulation experiments [16], the simultaneous simulation of holistic production systems, such as the brewery, has been made possible.

To represent batchwise production within the brewery (process areas: brewhouse, fermentation/storage/filter cellar) in the simulation environment, the following concepts of the necessary functionalities for the extension of the simulation system were developed and fully implemented. A fundamentally new development is the automatic simulation model generation based on the XML exchange file. The building blocks in the modeling editor are assigned to a newly developed batch building block within the simulation. The coordinate system and the links of the elements can be used to reformulate the logical structure of the plant. In contrast to the discrete domain, the batch module is based on a time-dependent specification of the individual Process Operations. This means that the Process Operations modeled in the process model are executed one after the other and the corresponding parameterized energy/media consumption level is applied for the respective duration of the Process Operation. The programmatic implementation in the batch module of the simulation environment follows the sequence "Fill—Process—Empty." At the runtime of the simulation, the corresponding parameters, such as the duration of the Process Operation and the corresponding consumption values, are loaded and reformulated if necessary and the simulation model is parameterized continuously. A special feature here is the integration of a production plan, i.e., the sequence of different recipes. This is mainly solved for the main source, the element in which material is

released in the simulation, by storing the non-production times and the times between the individual recipes and their specific article number. This indexing allows an element-specific parameter change and thus enables the recipe-specific simulation within the production plan. The simulation of each Process Cell takes place in its own interface within the simulation, but several interfaces can be active at the same runtime, thus enabling the holistic simulation of production systems in the beverage industry. To be able to analyze simulation tests in a flexible granularity and with regard to different sizes, the simulation results are stored in the presented database structure in an SQLite database. A fast and simple evaluation is carried out by a specially developed evaluation software in MATLAB 2020b AppDesigner [42], which allows a recipe- and plant-specific determination of the required energy and media quantities for each of the Process Operations.

The evaluation software connects to the results database and dynamically determines the respective process elements and units as well as the associated data points. A specific query code is generated to determine the consumption data within the respective time intervals of the Process Operations. The structure stored in the database makes it possible to compile the data into an overall picture. The procedure is implemented in user-friendly interfaces, through which the user is intuitively guided. The results are finally output in tabular form including all data.

## 7. Verification, Validation & Use Case Demonstration

The verification and validation of the method are intended to underpin the reliability of the model creation, the simulation and the validity of the simulation results and to serve as proof of the correct functioning. Especially with regard to possible resulting decisions in the implementation this is essential. The verification includes the examination of the conversion of a model such as the model in the modeling editor, into another model such as the simulation model. Therefore, the generated model was checked manually for correctness with regard to the topology, the elements and their correct connection. During the simulation run, the different parameter sets of the recipes were regularly checked and compared with the values within the model in the modeling editor. To check the results of the simulation and thus its correct functioning, a simulation run was evaluated manually using the freeware HeidiSQL_11 [48], which allows manual access to and viewing of SQLite databases, on the one hand and using the self-developed evaluation tool on the other.

Validation is to check the correct functioning of the model for the specific application purpose in the simulation compared to reality [49]. In the present case, the validation refers to the simulation of batch-oriented production in the brewhouse of a brewery. The measured and simulated consumption values are compared within the validation periods, according to Al-Hawari et al. [50], with the measured value representing the reference value.

The application of the combined simultaneous simulation of batch and discrete operation is to be demonstrated with the help of a use case. For this purpose, the method presented in this paper is combined with the method of the discrete packaging/bottling area of the beverage industry [16] and shown on the basis of the hybrid simulation of a brewhouse and a beverage bottling plant.

## 8. Results

### 8.1. Validation Periods and Simulation Parameter

Three different time periods, defined within the respective time periods of the measurement data recording, of the presented brewhouse are available for the validation of the simulation (see Table 1). In the first two measurement periods, different processes and thus units were examined and measured with regard to the electrical energy demand. In the third measurement period, the entire brewhouse and its processes were examined

with regard to thermal energy consumption. Each of the periods covered a different sequence of brews with a combination of three different recipes over a period of 140–170 h.

**Table 1.** Overview of the validation periods of the brewhouse including the brew cycle (W—Wheat, P—Pilsner, L—Lager).

| Validation Period | Duration/Number of Brews | Brew Cycle | Type of Energy/Media Investigated | Process Stages |
|---|---|---|---|---|
| 1 | 168.83 h/71 brews | 13× W, 19× P, 1× L, 18× W, 14× L, 1× P, 5× L | Electrical energy | Milling, Mashing, Lautering |
| 2 | 139.88 h/60 brews | 10× P, 1× L, 1× P, 1× W, 7× P, 10× L, 1× P, 9× L, 7× P, 13× W | Electrical energy | Heating up wort, Keeping heat, Vaporization & cooling |
| 3 | 143.9 h/60 brews | 17× L, 1× P, 18× W, 24× P | Thermal energy | Whole brewhouse process |

The number, times as well as energy and media quantities of the individual Process Operations were determined recipe-specifically. This was done semi-automatically by implementation in self-programmed software in MATLAB R2020b [42]. The start and end times of the Process Operations were derived manually and the time stamps were automatically transferred to the interval table of the database. For more precise delimitation, brew logs, step chain logs of the individual recipes and historical PLC data, such as valve cycles, were used. The software determines the resulting energy and media quantities of the Process Operations according to the method presented in chapter 5.

The simulation-relevant parameters were determined for measurement periods 1 and 2 in the following number for each recipe: 12× Wheat, 15× Lager, 15× Pilsner. When the thermal energy demand in measuring period 3 were recorded, 10 brews per recipe were evaluated. Thus, statistically validated simulation parameters of the individual Process Operations could be determined. Table A1 shows the Process Operations and the associated simulation parameters for the Pilsner recipe. In total, there are about 600 parameters for the entire simulation model with three recipes. The times between the individual brews (brew cycle) of the different recipes were determined as the mean on the basis of all the data and transferred to the model. The basis for this is the filling and startup of the malt mill.

*8.2. Model of the Brewhouse*

The brewhouse investigated in this work was modeled in the presented modeling editor. Figure 5a shows the physical model of the brewhouse consisting of the units as described in chapter 2. The special features of the modern boiling system include the pre-run tank, wort preheating (booster) and post-evaporation by the gentle boiling system. Therefore, the wort is kept hot in the wort kettle. The source serves as the interface to the upstream malt store and the sink as the interface to the fermentation tanks, corresponding to the transition to a possible further process cell, the cold block of the brewery. In the Wheat recipe, a decoction mashing process is used, i.e., the physical breakdown of the starch by boiling up a partial mash. A mash pan is used for this purpose. To simplify the model, the parallel process step of mash boiling was implemented energetically in the "Mashing" Process Stage in the mash tun. Figure 5b shows the process model on the Process Operation level within the Process Stage "Mashing" for the process of wort production for the Lager. The arrows define the respective sequences of the processes, respectively the linking of the units. The start and end elements of the process model represent the connection to the start and end of the respective higher Process Stage.

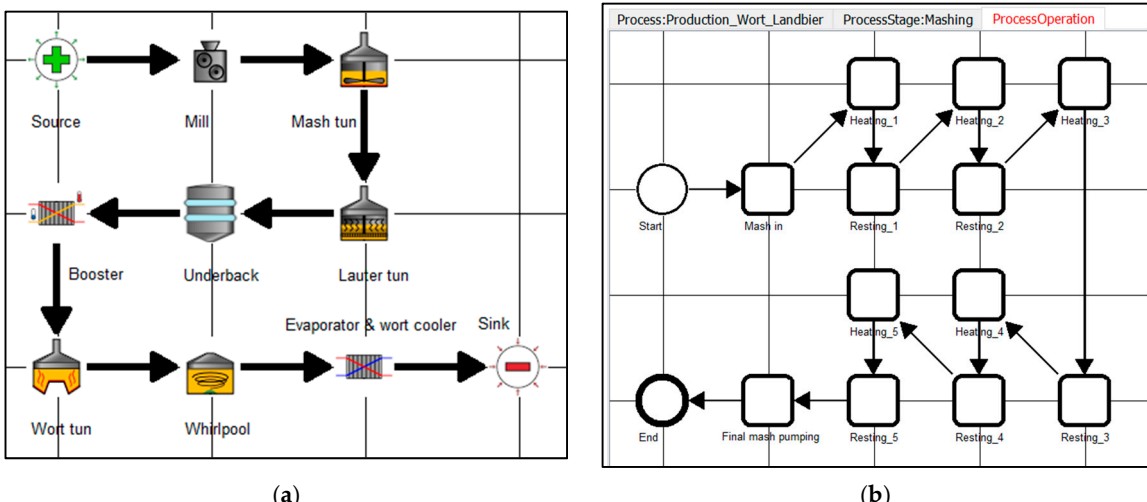

| (**a**) | (**b**) |

**Figure 5.** (**a**) Model of the presented brewhouse and (**b**) model of the Process Stage "Mashing" ("Lager") (adapted from [29]) in the modeling editor.

In the recipe model, the individual Process Stages within the process models were linked to the units of the physical model and additional information, such as the brew size, was indicated. The production plan was parameterized in each case as shown in Table 1. In addition, the times of the brewing sequence, the shift times and the respective start time of the simulation run were defined as the day of the week and the time of day.

### 8.3. Validation Results

#### 8.3.1. Model and Simulation Verification

As in the previous work of the authors [16], the verification of the automatic model generation in the simulation environment based on the XML configuration file resulting from the modeling editor was performed manually. In addition to verifying the correct generation of the simulation model, all parameters of all recipes were manually checked. In particular, the correct sequence of the individual Process Operations and their durations and energy/media consumption levels were examined. In addition, the sources, which represent the inflow of the main stream and thus the production schedule, were checked for correct functioning and production quantities and brew cycle times. The verification was performed continuously with the development process of the simulation environment and could be completed successfully.

#### 8.3.2. Validation of the Brewhouse Energy and Media Demand

The validation of the brewhouse electrical energy demand is divided into two periods, but in each case all processes were simulated with the complete parameter sets. This also applies to the validation of the thermal energy demand. In all three periods, the brew number is reached according to the periods in reality and a respective maximum time deviation of 1% could be determined. First, the curves of the electrical and thermal energy demand for one brew and for selected validation periods as well as detailed sections of them are presented. Subsequently, the validation and real values of the individual Process Stages as well as the entire periods are listed and their percentage deviation is shown.

Figure 6 shows the comparison of the simulated electrical consumption curve, which results from the determined parameters of the model, with the measured consumption values of a combined brew. Due to the division of the measurement exception of the electrical energy demand, the measured data curve consists of two different brews. It can be seen that the large consumption peaks caused by the pumps during "Mashing in" and the pumps in the Process Stages "Vaporization and Cooling" are reflected especially well.

Small consumption peaks, which sometimes only occur for a few seconds (see beginning), cannot be reproduced. The time offset results from the mean values of the Process Operation durations in the parameterized model.

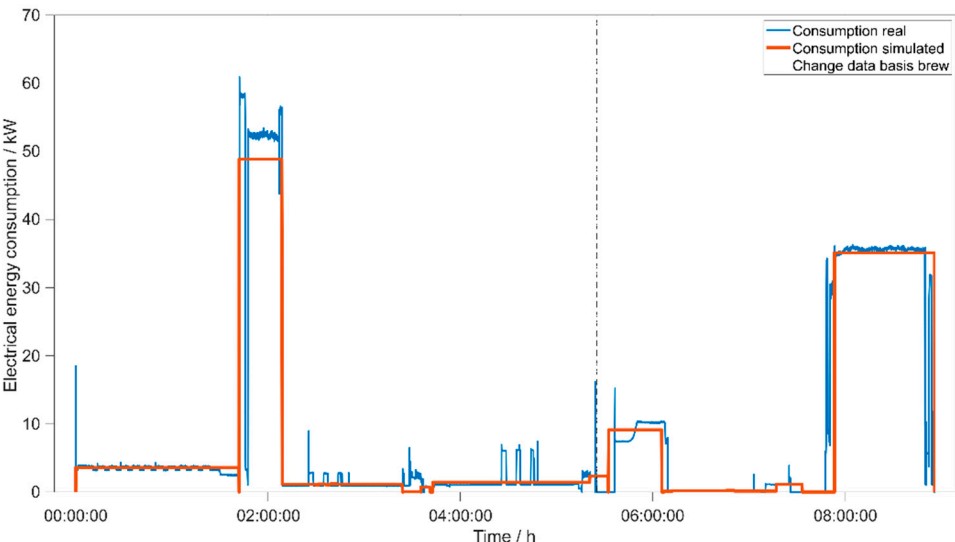

**Figure 6.** Comparison of the electrical energy consumption [kW] of one brew (real & simulated) "Lager".

Figure 7 shows the course of the electrical energy demand in validation period 1. The consumption curves of the individual processes and the cumulative total consumption are shown. The starting production at the beginning of the simulation, in which only the "Milling" and "Mashing" processes are initially active, can be seen. Since the lauter tun represents the bottleneck of the brewhouse, the times of the brew cycle are designed for the optimum utilization of this unit. As the simulation time progresses, the demand curves overlap more and more and several brews are active in different processes in the brewhouse at the same time. This sometimes results in very high demand peaks.

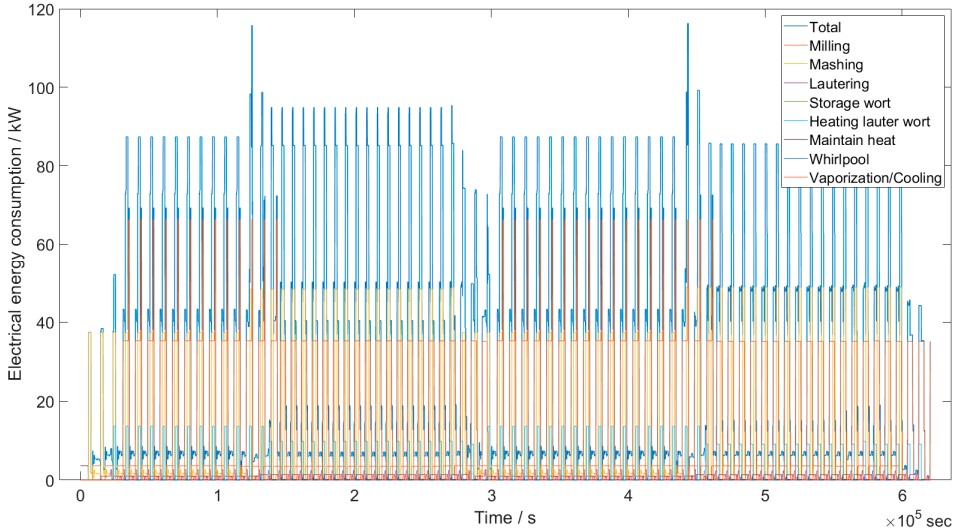

**Figure 7.** Electrical energy consumption [kW] of all processes of the brewhouse in validation period 1.

Figure 8 shows a detailed section with the different demand curves for electrical energy, which reflect the unit measured for a Process Stage or Process Operation. In addition, the associated Gantt chart for equipment occupancy on the unit level is

presented. Due to the parallelism of processes such as simultaneous "Mashing" (e.g., pump during mashing and pump during boosting) and "Vaporization and Cooling", enormous consumption peaks sometimes occur. This also causes the two extremes, in which both Process Stages of different recipes overlap (see Figure 7).

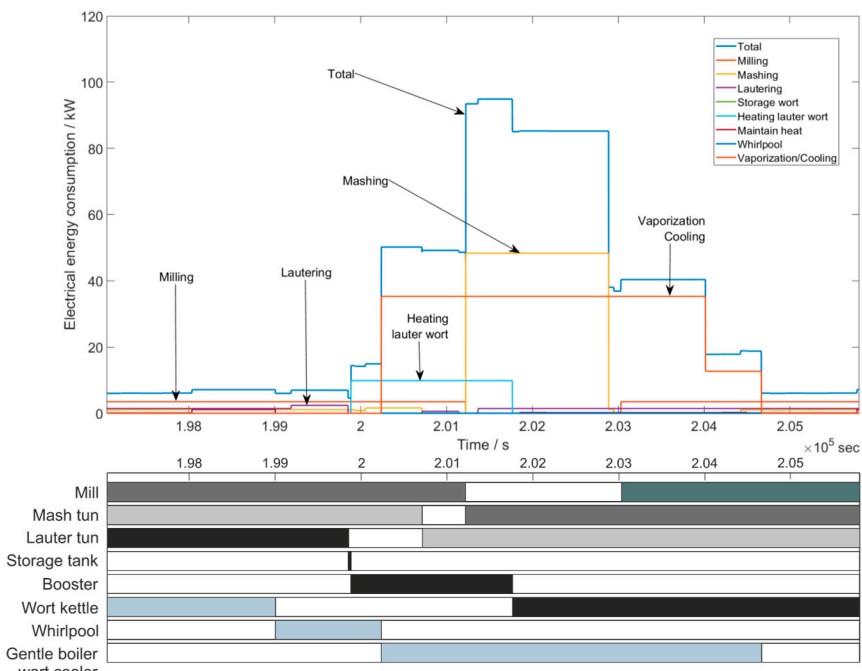

**Figure 8.** Detailed electrical energy consumption [kW] of all processes of the brewhouse in validation period 1 incl. the equipment occupancy in a Gantt chart.

The validation of brewhouse thermal energy consumption is based on the continuous measurement of the individual units within a period of time. Figure 9 shows the comparison of the real measured and simulated consumption of thermal energy during one brew. The times of the Process Operations and the consumption parameters of the simulation are based on the mean values from the data acquisition. The two curves agree well, although short peaks, sometimes lasting only a few seconds, cannot be accurately represented by averaging the real consumption curve.

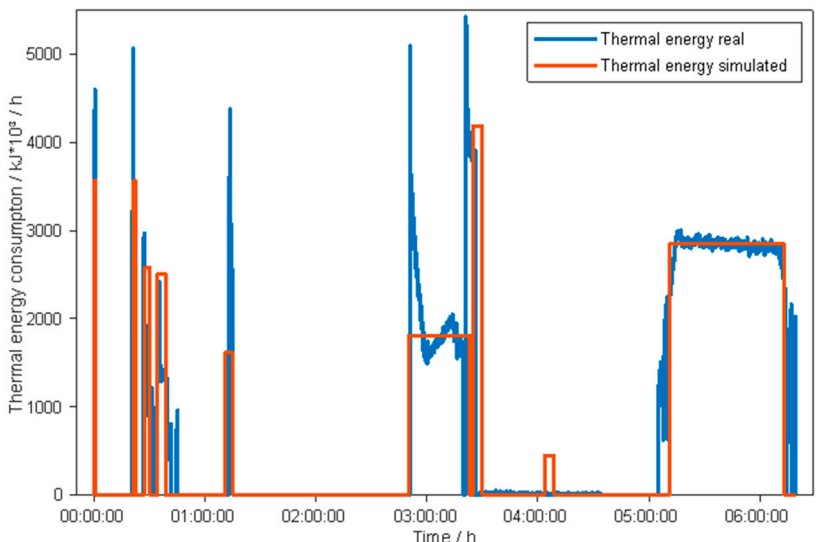

**Figure 9.** Comparison of thermal energy demand [kJ/h] of one brew (real & simulated) "Lager".

Figure 10 shows the course of the thermal energy demand in validation period 1. The consumption curves of the individual Process Stages and the cumulative total consumption are listed. No thermal energy is required for the Process Stage "Vaporization and Cooling", but is generated recuperatively via heat exchangers. This energy is reused within the production system. The constantly repeating consumption profiles, depending on the recipe, can be seen.

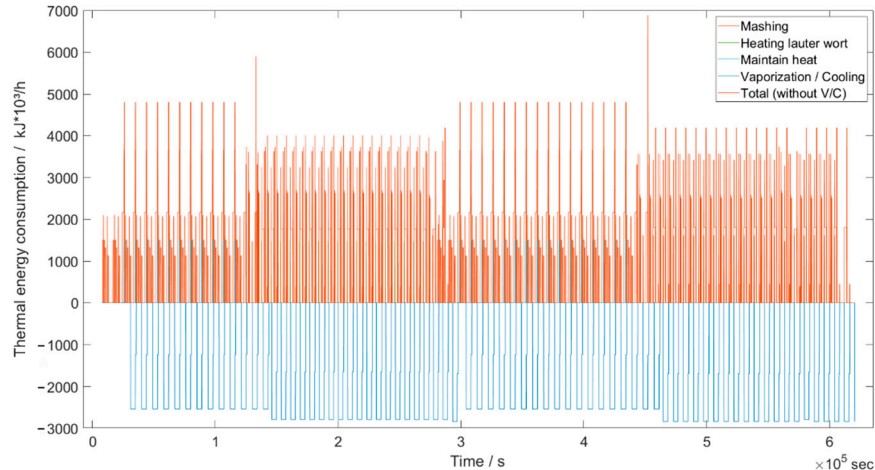

**Figure 10.** Thermal energy consumption [kJ/h] of all processes of the brewhouse in validation period 1.

Figure 11 shows a section of the thermal energy demand as well as the associated equipment occupancy chart in detail. The individual process days and the Process Operations in which thermal energy is required are marked. A total demand curve is also shown. The "Vaporization and Cooling" Process Stage is not included in the overall analysis.

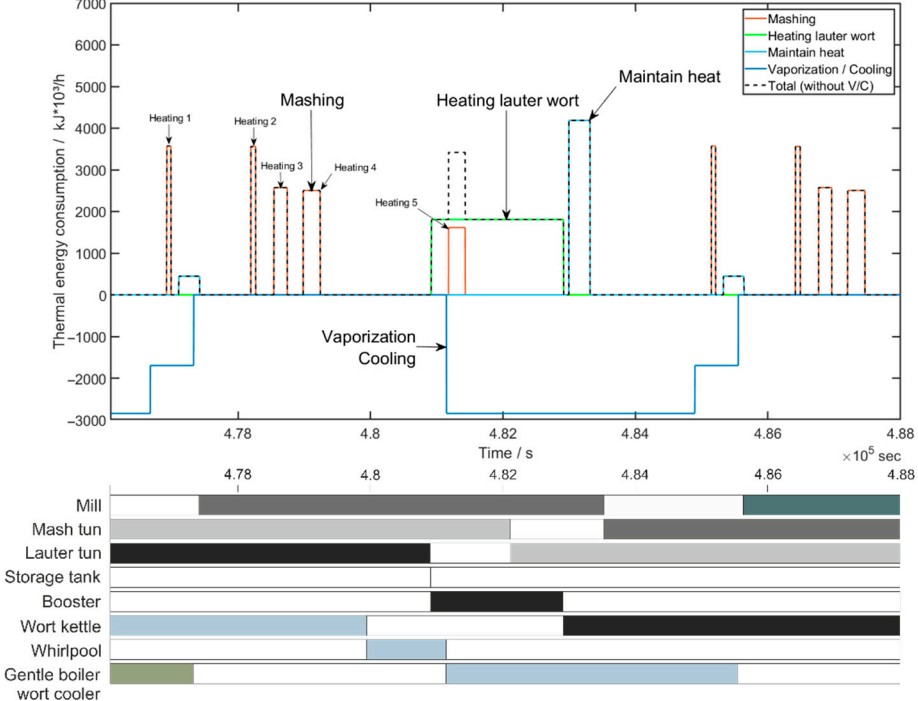

**Figure 11.** Detailed thermal energy consumption [kJ/h] of all processes of the brewhouse in validation period 1 incl. the equipment occupancy in a Gantt chart.

Table 2 shows the validation of the electrical and thermal energy demand by the comparison of the simulated and measured consumption quantities and their percentage deviation. The validation is carried out generally and specifically according to the division of the Process Stages and the respective recipe. Due to the divided measurement recordings, the respective values refer to the corresponding measurement and validation periods. The values also differ in the number of brews, the recipes and the sequence. The two Process Stages "Wort Storage" in the underback and "Separation" in the whirlpool are not listed in the table, as no energy or media consumption was assigned to these steps. The values were determined by one simulation run per time period since, due to the lack of stochasticity of the duration of the individual Process Operations, a multiple simulation would not show any deviating consumption values.

**Table 2.** Validation results for electrical and thermal energy demand (simulated, measured & percentage deviation) for all Process Stages and for all recipes in the three validation periods.

| Validation Period | Unit | Process Stage | Electrical Energy [kWh] | | | | | | | | | | | |
|---|---|---|---|---|---|---|---|---|---|---|---|---|---|---|
| | | | Lager | | | Pilsner | | | Wheat | | | Total | | |
| | | | Sim. | Meas. | PD [%] | Sim. | Meas. | PD [%] | Sim. | Meas. | PD [%] | Sim. | Meas. | PD [%] |
| 1 (20× L; 20× P; 31× W) | Mill | Milling | 121.94 | 119.61 | 1.95 | 115.05 | 115.59 | −0.46 | 181.33 | 176.88 | 2.51 | 418.32 | 412.08 | 1.51 |
| | Mash tun/pun | Mashing | 467.59 | 475.44 | −1.65 | 472.61 | 500.91 | −5.65 | 724.16 | 753.63 | −3.91 | 1664.36 | 1729.98 | −3.79 |
| | Lauter tun | Lautering | 57.02 | 56.88 | 0.25 | 54.38 | 54.96 | −1.05 | 81.79 | 81.20 | 0.74 | 193.19 | 193.03 | 0.08 |
| 2 (20× L; 26× P; 14× W) | Booster | Heating up wort | 101.13 | 100.03 | 1.10 | 133.53 | 124.12 | 7.58 | 77.50 | 76.54 | 1.25 | 312.17 | 300.70 | 3.81 |
| | Wort kettle | Keeping heat | 10.08 | 8.71 | 15.71 | 12.44 | 11.86 | 4.85 | 5.32 | 5.80 | −8.23 | 27.84 | 26.37 | 5.56 |
| | Gentle boiler, wort cooler | Vaporization & cooling | 776.22 | 738.60 | 5.09 | 1021.32 | 970.50 | 5.24 | 539.42 | 542.93 | −0.65 | 2336.96 | 2252.03 | 3.77 |
| | Brewhouse | Total | 1533.98 | 1499.27 | 2.32 | 1809.34 | 1777.95 | 1.77 | 1609.52 | 1636.98 | −1.68 | 4952.84 | 4914.20 | 0.79 |

| Validation Period | Unit | Process Stage | Thermal Energy [kJ] | | | | | | | | | | | |
|---|---|---|---|---|---|---|---|---|---|---|---|---|---|---|
| | | | Lager | | | Pilsner | | | Wheat | | | Total | | |
| | | | Sim. | Meas. | PD [%] | Sim. | Meas. | PD [%] | Sim. | Meas. | PD [%] | Sim. | Meas. | PD [%] |
| 3 (17× L; 25× P; 18× W) | Mash tun/pun | Mashing | $9.73 \times 10^6$ | $9.10 \times 10^6$ | 6.98 | $1.44 \times 10^7$ | $1.38 \times 10^7$ | 4.62 | $2.36 \times 10^7$ | $2.22 \times 10^7$ | 6.55 | $4.78 \times 10^7$ | $4.50 \times 10^7$ | 6.04 |
| | Booster | Heating up wort | $1.70 \times 10^7$ | $1.56 \times 10^7$ | 8.93 | $2.31 \times 10^7$ | $2.38 \times 10^7$ | −2.89 | $1.58 \times 10^7$ | $1.55 \times 10^7$ | 1.73 | $5.59 \times 10^7$ | $5.49 \times 10^7$ | 1.78 |
| | Wort kettle | Keeping heat | $5.44 \times 10^6$ | $5.45 \times 10^6$ | −0.15 | $7.89 \times 10^6$ | $7.89 \times 10^6$ | 0.01 | $6.42 \times 10^6$ | $6.05 \times 10^6$ | 6.13 | $1.97 \times 10^7$ | $1.94 \times 10^7$ | 1.87 |
| | Gentle boiler, wort cooler | Vaporization & cooling | $5.56 \times 10^7$ | $5.26 \times 10^7$ | 5.65 | $8.08 \times 10^7$ | $7.98 \times 10^7$ | 1.16 | $5.41 \times 10^7$ | $5.54 \times 10^7$ | −2.30 | $1.90 \times 10^8$ | $1.88 \times 10^8$ | 1.40 |
| | Brewhouse | Total | $8.78 \times 10^7$ | $8.28 \times 10^7$ | 6.03 | $1.26 \times 10^8$ | $1.25 \times 10^8$ | 0.70 | $9.99 \times 10^7$ | $9.91 \times 10^7$ | 0.82 | $3.14 \times 10^8$ | $3.07 \times 10^8$ | 2.18 |

In the area of electrical energy, a very small deviation of 0.8% can be determined in the overview. The largest deviation concerns the process step "Keeping heat" within the wort kettle. However, this Process Stage has a small share of the total demand, 0.56%. The deviations are also reflected in the recipe-specific analysis. Furthermore, the steps "Mashing", "Heating up Wort" and "Vaporization & Cooling" show similar deviations in the range of 3.8%. In these processes, electrical energy is required mainly for pumps to transfer the brew, which covers a similar volume across the recipes. Some of the largest deviations in the values from validation period 2 are achieved for the Pilsner recipe. This is due to the relationship between the basis of the number of brews measured in the measurement period, which vary in the range of electrical energy, versus the number of brews in the validation period. In general, it can be concluded here that a larger number

of brews examined in the parameter determination leads to more exact agreements within the validation periods. Nevertheless, very small deviations between the measured and simulated values can be observed uniformly in relation to the respective recipes.

In the area of thermal energy, the largest deviations exist for the "Mashing" Process Stage. This is clearly due to the high number of Process Operations (12–14) in the mashing and the associated inaccuracy in parameter determination due to the definition of the time limits. Especially in the Lager recipe, high deviations occur in the "Heating wort" step due to the booster. In combination with the deviations in vaporization/cooling, this results in a total deviation of 6% for this recipe. This phenomenon is due to the low database for "Vaporization & Cooling", as there were also difficulties with the measurement during operation. The Process Stage "Keeping heat" also shows high deviations for the recipe Wheat, but accounts for only 6% of the total thermal energy requirement. The remaining simulated values agree very well with the measured values and a maximum deviation of 2% is achieved in each case. Overall, this results in very small deviations for the Pilsner and Wheat recipes and a total percentage deviation of 2.2% is achieved.

### 8.3.3. Use Case

In a previous study, a use case was developed for the simulation of a brewhouse with regard to the reduction of peak loads of electrical energy by adjusting the brew cycle times. A reduction in peak loads of up to 32% was achieved [29].

For the first time in the scientific literature, the holistic hybrid simulation of batch-oriented and discrete operation is presented. For this purpose, the presented method for the simulation of the batch area of the brewery is used in combination with the already presented approach for the simulation of packaging and bottling plants [16]. The presented brewhouse and a returnable glass beverage bottling plant were modeled together in the modeling editor. The simulation model was automatically generated in the simulation environment based on the XML configuration file.

The resulting model comprises two process cells (batch-oriented brewhouse and discrete beverage bottling plant) with three recipes and four filling articles. All parameters were evaluated on a recipe/article-specific basis and the model was parameterized accordingly. In addition, corresponding shift plan was defined for both areas to define the daily start and end times. Production in the brewhouse is round the clock, whereas in the bottling area production takes place from Monday 7:00 am to Saturday 8:30 am. Toward the end of the simulation period, a weekend is therefore depicted. In addition, the brew cycle times and changeover times between the filling articles are stored for both process cells. The simulation covers a total time of 7.5 days, starting at Sunday 10 pm. Figure 12 shows the course of the electrical energy demand of the brewhouse and of the beverage bottling plant and the sum of these. The delayed start-up of the beverage bottling plant due to the limited shift schedule for this process cell can be seen clearly. Furthermore, the resulting total consumption and the resulting peaks due to the overlapping of many individual electrical energy consumers can be seen well.

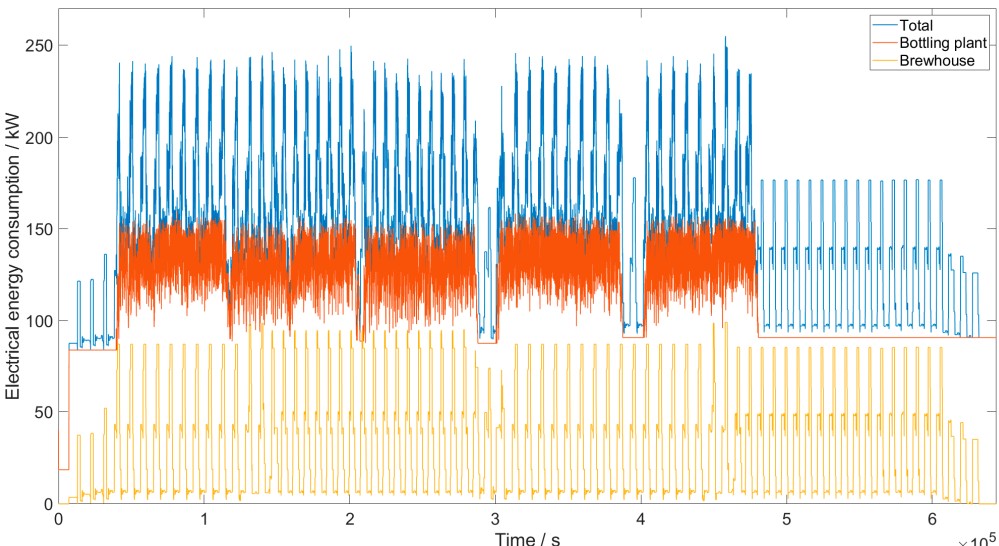

**Figure 12.** Electrical energy consumption of the combined simulation of the brewhouse and the beverage bottling plant.

The use case shows that the entire method is suitable for a holistic analysis and forecast of the energy and media consumption of plants in the beverage industry and can simulate any number of plants of the batch-oriented as well as the discrete operation mode. The method enables holistic energetic consideration and opens numerous possibilities for carrying out experimental tests to increase energy and media efficiency on the basis of numerous factors.

## 9. Discussion

Due to the structured subdivision of plants and the corresponding processes according to ANSI/ISA-88 [39], the data acquisition procedure is well suited for transferability to any batch-oriented production operation. Although standardized procedures for collecting data on energy and media consumption are available, such as for packaging machines [51], no approach has yet been described for the batch-oriented production area of the brewery. Since the collection of data is very time-consuming, a solution is especially needed for the application of a standardized determination of consumption parameters.

Due to the flexible subdivision of the Process Operations, energy and media consumption behavior can be mapped in any granularity. The effort of manually defining the start and end times of the Process Operations has optimization potential. This effort can be automated if reliable raw data are available, for example relating to the sequence of the step chain. Also conceivable is the automatic detection of peaks, although the definition of a peak and the relevance of a peak remain open. Nevertheless, the one-time manual delimitation of Process Operations on the basis of consumption curves and with the help of context data such as brew logs, step chain logs and other PLC data make it possible to precisely delineate the individual Process Operations. On the basis of the standardized data structure, the simulation-relevant, recipe-specific consumption parameters can be calculated automatically in a standardized way using a self-developed tool in MATLAB R2020b [42]. This enables the comparability of the simulation parameters. The parameters consistently show very low confidence intervals and can therefore be considered reliable. The parameter determination tool is structured via graphical interfaces and can thus be used without specific prior knowledge or requirements.

The model of the brewhouse with all parameters and the production plan model were created in a structured and standardized manner in the graphical modeling editor,

which is based on modeling columns. Due to its ease of use, the model can be created quickly and checked for correctness with the help of help functions and automatic checks. Compared to time-consuming manual parameter determination and simulation model creation, which in this case require several weeks, the present simulation model can be created within a few days with the help of the parameter determination, the modeling editor and the automatic simulation model generation. The prerequisite for this is a reliable database. Fast model creation overcomes a significant barrier for companies in the beverage industry. A novelty is the integration of a production plan with different recipes, which are represented by different parameter sets. The automatic generation of simulation models of the batch-oriented mode of operation can be shown on the basis of the XML configuration file as exemplified by the used simulation environment. Since the configuration file contains all simulation-relevant information, the automatic model generation is transferable to any simulation environment.

Based on three different time periods of the brewhouse, an extensive validation can be carried out to prove the suitability of the entire method. The simulated values were compared with the real measured values and the percentage deviation was determined. The evaluation was carried out both overall and recipe-specific. The validation is limited to the electrical and thermal energy consumption of the brewhouse of a brewery as no measured data were collected for other types of energy and media or other areas such as fermentation/storage cellar and filtration. In all periods, the production plan was processed as specified and the simulated time deviates only very slightly from the production time required in reality.

The direct comparison between the simulated course of a brew and a real brew shows well the agreement of the temporal and quantitative demand of electrical and thermal energy. The values of the simulated course result from the mean values of the determined times and consumption parameters of the individual Process Operations and are therefore repeated for each recipe. This explains the time deviations due to the slight shifts of individual consumption peaks in both figures. There are also individual peaks which are not represented by Process Operations and are therefore disregarded. Most of these peaks occur only very briefly (a few seconds) and also have only a minor effect on the total electrical energy demand. In the area of thermal energy, the short maximum peaks are only partially mapped due to the averaging of the consumption value over the duration. The figures of the detailed electrical and thermal energy curves of the individual process stages, together with the unit occupancy Gantt charts, show well how the consumption peaks are formed. In reality the occupancy of two consecutive units overlaps due to the simultaneous emptying and filling of the vessels. In the simulation this is simplified by a direct transition. In the brewhouse, several brews are usually active because of the bottleneck, the lauter tun, which leads to a simultaneity of Process Stages. Especially when different recipes follow one another, very high consumption peaks can sometimes occur due to overlapping. A use case already presented deals with the shifting of consumption peaks by adjusting the times of the brew cycle in the production plan [16]. The overall view of the consumption curves in validation period 1 illustrates this. The functionality of the simulation is clarified by the representation of the complex process structure, which is significantly influenced by the production plan. This offers a good opportunity for companies in the beverage industry to depict the total demand of their production facilities and to test possible efficiency improvements. The thermal energy curve in particular illustrates how much thermal energy results in the brewing process and can be further used as recuperative energy. This is state of the art and is discussed in numerous publications in the form of sources and sinks as well as heat cycles and is not the subject of further investigation in this work.

The numerical comparison of the validation is performed in the granularity of the Process Stages and the recipes. The three validation periods have a similar duration, different sequences of the brews and a varying number of brews. The brews, which serve as the database for parameter determination, are outside the validation periods. This

ensures that the method of parameter determination is also included in the validation. The total values refer to the sum of the Process Stages within the respective recipe due to the split measurement periods. Overall, a very good agreement of the electrical energy demand is achieved. Most of the consumptions concerning the Process Stage "Keeping heat" can be disregarded due to their small share of the total demand. The "Vaporization & Cooling" Process Stages, together with "Mashing", have the greatest influence on the total requirement and also sometimes show the highest deviations. This goes hand in hand with the number of Process Operations within these Process Stages. It can therefore be concluded that, in conjunction with the number of brews investigated, the division into Process Operations has a significant influence on the deviations. A larger database as well as the automated recognition of the Process Operations would remedy this situation. This also applies to the deviations of the thermal energy values. Especially for heating processes, a small database is a challenge, since numerous external influences can affect the required energy. Nevertheless, very small deviations are achieved overall and for the Pilsner and Wheat recipes.

A novelty describes the combined simulation of the batch-oriented and the discrete mode of operation in the use case. The challenge here is the simultaneous representation of the time-based batch process with changing energy and media levels depending on the Process Operations and the stochastic state change-based energy and media levels of the units in the packaging/bottling area. The use case represents the brewhouse and a refillable glass beverage bottling plant and thus provides a good example of a beverage industry operation. The method is not limited in terms of the number or type of process cells. Another special feature is the integration of specific production schedules. This integration allows the realistic simulation of the real production and thus extends the scope for experimental investigations by adapting them. As a result, the overall course of the electrical and thermal energy for the use case is presented. Only the holistic view of a production operation allows the investigation of the influence on the entire system. This is essential especially for the avoidance of load peaks, for design planning of supply facilities, for the generation and use of recuperative energies and finally for the increase in energy and media efficiency by experimental investigations based on numerous influencing factors.

## 10. Conclusions

An approach for the modeling and simulation of batch-oriented operation with regard to energy and media requirements of plants of the beverage industry was presented. By using a previously presented modeling solution and by the development of a semi-automatic parameter determination tool, a model of a brewhouse can be created. In order to obtain a reliable database for the determination of the simulation parameters on the one hand and for the acquisition of validation data on the other hand, extensive measurement campaigns regarding brewhouse electrical and thermal energy demand were carried out. By using a configuration file, a simulation model can be automatically generated in a simulation environment and successfully verified. Based on three time periods, the entire method was successfully validated. Very small deviations were achieved throughout for the total and recipe-specific thermal and electrical energy consumption values. It can be concluded that the simulation method is very well suited to depicting and predicting the energy and media consumption behavior of process plants in the beverage industry in detail. The number of Process Operations within Process Stages influences accuracy immensely, which indicates a potential for improvement in the definition of these operations. This can be done by automated detection of the consumption-relevant Process Operations. In addition, the database plays an essential role for the accuracy of the energy and media consumption quantities within the time periods. Although an extensive database could be created, further data is advantageous, especially for a recipe-specific consideration. The automated storage of consumption data would eliminate the need for time-consuming measurement campaigns, data preparation

and manual linking with contextual information. In particular, the presented database structure in connection with the nomenclature of the Weihenstephan Standards [43] offer a good basis for an automated connection of a process control system. In addition to the direct specification of the respective consumptions of the processes by the respective plant manufacturer, numerous advantages would result, in particular with regard to the design planning of the supply plants and the subsequent examination of optimization measures.

The claim to create a user-friendly method that can be used without specific previous knowledge or expert know-how in order to enable SMEs above all to increase their energy and media efficiency is guaranteed with the tools for automatic parameter determination, simulation evaluation and by the modeling solution including automatic simulation model generation. In particular, the modeling solution has been described and validated in detail [29].

The use case for simultaneous simulation of batch-oriented and discrete production mode in the brewery forms the overall picture from the previous work on the model for mapping holistic production systems of the beverage industry. This includes the implementation of the metamodel in a modeling editor [29] and automatic model generation and the simulation of the discrete packaging/filling area of beverage plants [16]. Furthermore, their specific recipes or articles can be mapped and simulated in a model with the inclusion of a production plan, which ensures a realistic simulation over even a longer period of time.

- The following scenarios for real use cases can be set up for which we believe the entire method is capable of solving: Prediction of realistic energy and media consumption of plants and machines.
  - For the correct design of utilities, since usually no or only very inaccurate information about the real consumption is available.
  - To detect unnecessary consumption, e.g., during downtimes and to apply consumption-oriented control.
  - To identify load peaks in certain time intervals to reduce and shift them in order to avoid overruns in the averaged power interval.
- Estimation and quantification of investments and their impact on the overall system.
  - Modernization of entire areas and individual units.
  - Capacity increases including any necessary adjustments to the supply structure.
  - Use of alternative sustainable energy generation systems (e.g., CHP plants, photovoltaic systems, etc.).
  - Use of energy and heat storage systems.
- Use of recuperative energies as well as waste heat in the entire production system.
- Adaptation of production plans for optimal energy and media utilization.
- Influence of factors and key figures (OEE, availability, etc.) on energy and media consumption to determine the optimal sustainable operation of the plants.

All application cases can contribute to more efficient and thus more sustainable production with the advantage that ongoing production is not affected or has to be interrupted. In addition, the entire method, through strict adherence to and implementation of ANSI/ISA 88 [39] structuring, shows potential transferability to other industries, such as the food, biopharmaceutical and chemical industries.

**Author Contributions:** Conceptualization, R.M.B.; methodology, R.M.B., S.S. and M.Z.; software, R.M.B., M.Z., J.K. and K.B.; validation, R.M.B.; formal analysis, R.M.B., S.S. and M.Z.; investigation, R.M.B. and S.S.; resources, J.K. and K.B.; data curation, R.M.B., S.S. and M.Z.; writing—original draft preparation, R.M.B. and S.S.; writing—review and editing, K.G. and T.V.; visualization, R.M.B. and M.Z.; supervision, K.G. and T.V.; project administration, R.M.B.; funding acquisition, K.G. and T.V. All authors have read and agreed to the published version of the manuscript.

**Funding:** This research was funded by the Bayerische Forschungsstiftung (AZ-1217-16).

**Institutional Review Board Statement:** Not applicable.

**Informed Consent Statement:** Not applicable.

**Data Availability Statement:** Data not available due to legal restrictions.

**Conflicts of Interest:** The authors declare no conflict of interest.

## Appendix A

**Table A1.** Overview of the simulation-relevant parameters of the "Pilsner" recipe.

| Process Stage | Process Operation | Duration | | Electrical Energy Demand [kWh] | | Thermal Energy Demand [kJ] | |
|---|---|---|---|---|---|---|---|
| | | MW | 95% KI | MW | 95% KI | MW | 95% KI |
| Milling | Milling | 5932 | 181.585 | 5.744 | 0.200 | - | -x |
| Mashing | Mashing in | 1663 | 56.100 | 22.334 | 1.050 | - | - |
| | Heating up 1 | 61 | 6.303 | 0.0181 | | 63,242.47 | 7275.96 |
| | Resting 1 | 1474 | 2.451 | 0.437 | | - | - |
| | Heating up 2 | 71 | 9.201 | 0.021 | | 71,164.83 | 3455.64 |
| | Resting 2 | 273 | 5.158 | 0.081 | 0.047 with a | - | - |
| | Heating up 3 | 185 | 34.982 | 0.055 | summed | 138,320.35 | 4138.56 |
| | Resting 3 | 238 | 19.993 | 0.071 | mean value | - | - |
| | Heating up 4 | 247 | 41.998 | 0.073 | of 1.457 | 180,228.15 | 2328.48 |
| | Resting 4 | 1935 | 10.747 | 0.574 | | - | - |
| | Heating up 5 | 302 | 38.145 | 0.090 | | 134,430.97 | 1939.32 |
| | Resting 5 | 125 | 0.350 | 0.037 | | - | - |
| | Final mash pumping | 656 | 6.429 | 0.294 | 0.001 | - | - |
| Lautering | Trub wort pumping | 429 | 123.694 | 0.07 | 0.070 | - | - |
| | Resting | 223 | 8.326 | - | - | - | - |
| | Lautering | 5566 | 51.070 | 2.20 | 0.220 | - | - |
| | Remove spent grains | 661 | 7.841 | 0.44 | 0.450 | - | - |
| Wort storage | Wort storage | 32 | 1.469 | - | - | - | - |
| Lauter wort heating | Lauter wort heating | 1877 | 54.028 | 4.780 | 0.138 | 923,352.70 | 32,161.68 |
| Keeping heat | Resting 1 | 80 | 1.001 | 0.003 | 0.003 with a summed mean value of 0.186 | - | - |
| | Heating up 1 | 324 | 23.917 | 0.020 | | 285,960.00 | 34,153.56 |
| | Resting 2 | 2022 | 326.276 | 0.103 | | - | - |
| | Heating up 2 | 306 | 12.938 | 0.016 | | 29,480.00 | 3773.16 |
| | Resting 3 | 1279 | 374.267 | 0.044 | | - | - |
| | Casting | 983 | 7.677 | 0.295 | 0.297 | - | - |
| Sedimentation | Sedimenting | 1230 | 64.335 | - | - | - | - |
| Vaporization and Cooling | Vaporization & Cooling | 3774 | 24.021 | 36.983 | 0.403 | −2,930,540.80 | 42,980.04 |
| | Rinsing | 650 | 34.496 | 2.294 | 0.033 | −298,660.00 | 3594.93 |

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
