# Peer review of "Simulation of Energy and Media Demand of Batch-Oriented Production Systems in the Beverage Industry"

_sustainability, doi:10.3390/su14031599_

Round 1
Reviewer 1 Report
Include a couple of Gantt and Equipment Occupancy charts along with the resource charts so that the readers can better visualize the operation of the brewhouse.
Reviewer 2 Report
The authors describe a rather interesting approach, which aims to optimize the batch-oriented production processes in the industry of beverages.
I would suggest that the authors describe, with more details, the data collection hardware, and the data processing infrastructure, both from a software and hardware perspective.
Furthermore, clearer statements concerning the usefulness of this approach in real-world scenarios should be included.
The English language should be entirely proofread and improved.
